# Distance-Aware Non-IID Federated Learning for Generalization and Personalization in Medical Imaging Segmentation

**Julia Alekseenko**[1,2]                                    JULIA.ALEKSEENKO@EXT.IHU-STRASBOURG.EU

**Alexandros Karargyris**[1]                                   AKARARGYRIS@GMAIL.COM

**Nicolas Padoy**[1,2]                                        NPADOY@UNISTRA.FR

[1] *IHU Strasbourg, Institute of Image-Guided Surgery, Strasbourg, France*

[2] *University of Strasbourg, CNRS, INSERM, ICube, UMR7357, Strasbourg, France*

**Editors:** Accepted for publication at MIDL 2024

## Abstract

Federated learning (FL) in healthcare suffers from non-identically distributed (non-IID) data, impacting model convergence and performance. While existing solutions for the non-IID problem often do not quantify the degree of non-IID nature between clients in the federation, assessing it can improve training experiences and outcomes, particularly in real-world scenarios with unfamiliar datasets. The paper presents a practical non-IID assessment methodology for a medical segmentation problem, highlighting its significance in medical FL. We propose a simple yet effective solution that utilizes distance measurements in the embedding space of medical images and statistical measurements calculated over their metadata. Our method, designed for medical imaging and integrated into federated averaging, improves model generalization by downgrading the contribution from the most distant client, treating it as an outlier. Additionally, it enhances model personalization by introducing distance-based clustering of clients. To the best of our knowledge, this method is the first to use distance-based techniques for providing a practical solution to the non-IID problem within the medical imaging FL domain. Furthermore, we validate our approach on three public FL imaging radiology datasets (FeTS (Pati et al., 2021), Prostate (Liu et al., 2020b), (Liu et al., 2020a), and Fed-KITS2019 (Terrail et al., 2022)) to demonstrate its effectiveness across various radiology imaging scenarios.

**Keywords:** Federated Learning, Non-IID Data, Personalization, Generalization, Medical Segmentation, Medical Imaging.

## 1. Introduction

Federated learning (FL) in healthcare aims to achieve data collaboration while preserving privacy. It enables multiple institutions or healthcare entities (clients) to jointly train or evaluate artificial intelligence (AI) models without sharing raw, sensitive patient data. This collaborative and privacy-preserving approach improves predictive models, personalized treatments, and disease detection, leveraging diverse datasets from various institutions and democratizing the power of distributed clients. FL promotes inclusive model training, incorporating diverse populations for robustness and generalizability. Studies like (Sheller et al., 2020; Dou et al., 2021) highlight FL's efficacy in medical applications, showcasing the power of algorithms like federated averaging (FedAvg) (McMahan et al., 2017).

While traditional FL algorithms, like FedAvg, assume uniform data distribution across clients, in contrast, real-world applications often face non-independently and identically

distributed (non-IID) data challenges, where data across clients lacks uniformity. Factors like disease manifestation, patient populations or image acquiring protocols contribute to this heterogeneity, impacting model convergence and performance (McMahan et al., 2017). Recent approaches on improving the generalization of the global model, such as FedProx by Li et al. (Li et al., 2020), regulate local updates to improve model generalization, solutions like FedBN (Li et al., 2021b) and FedCross (Xu et al., 2022) address non-IID scenarios by optimizing feature spaces or sequentially training the global model across clients. Another essential strategy in FL is personalization, which involves training a specific model for each client while leveraging insights from others. Recent advancements in personalized FL include training one model per participating institution through adaptations of meta-learning (Fallah et al., 2020; Acar et al., 2021), multi-task learning (Marfoq et al., 2021), utilizing partial model sharing (Pillutla et al., 2022), local fine-tuning (Li et al., 2021a; Yu et al., 2020), and clustering solutions (Ghosh et al., 2022; Manthe et al., 2023).

However, the majority of proposed works aim to accept client data distributions as non-IID without measuring the heterogeneity of the federation and integrating this information into the pipeline. Assessing non-IID characteristics can provide crucial insights into training challenges and generalizability, for instance, (Zhao et al., 2018) observed decreased accuracy in federated models with higher Earth Mover's Distance (EMD) in non-IID image datasets. Yet, their study only evaluated classification problems with basic datasets like MNIST and CIFAR-10, limiting its applicability to healthcare. In the medical domain, (Luo et al., 2023) proposed analyzing data distributions related to site, tumor type, tumor size, dataset size, and tumor intensity. They demonstrated a significant negative correlation between the Dice score ratio and data distribution distances, particularly with EMD, in medical image segmentation. However, practical solutions for improving the federation based on these findings have not been proposed.

Our main contribution lies in integrating EMD distance-related insights into federated averaging to offer optimal training for both generalization and personalization strategies, surpassing the performance of traditional FL methods. Assuming that this approach may not capture all non-IID characteristics, we explore non-IID measurements within the embedding space of data (i.e., medical images). We utilize a publicly-available pre-trained model to extract rich and meaningful embeddings, and then calculate the Euclidean distance (EUC) based on them. To the best of our knowledge, this is the first work in the medical imaging FL to propose this concept.

## 2. Method

Our methodology, presented in Figure 1, aims to achieve two important yet opposing goals in FL: a) model generalization, focusing on improving the accuracy and generalization of the model, and b) model personalization, tailoring the model for the highest accuracy at the client level. Our proposed methodology can be subdivided into two (2) steps. The first step involves measuring the degree of data heterogeneity (non-IID) among clients. To achieve this, our methodology runs two approaches in parallel. The first approach relies on statistical distances (i.e., EMD) based on metadata from medical images (Subsection 2.1), while the second one calculates EUC in the embedding space of these images (Subsection 2.2). For this, a publicly pre-trained model is deployed from the server to each client

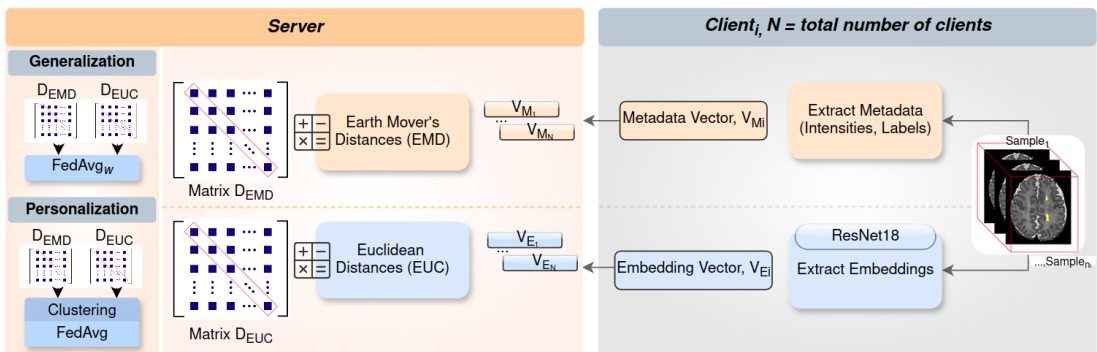

Figure 1: Proposed method for medical FL optimization.

for extracting embeddings. Subsequently, we extend the federated averaging algorithm by incorporating the down-weighting of the most distant client to enhance generalization. We also augment the personalization strategy by clustering closely-related clients (Subsection 2.3). While we hypothesize that the federation is trusted, we recognize the potential for privacy enhancement. However, the investigation into the compatibility of our methodology with privacy-preserving techniques (Jin et al., 2023), (Wei et al., 2020) is beyond the scope of this study.

## 2.1. Non-IID Assessment with EMD Statistical Distance

To assess the non-IID nature of the federation, characterized by disparities between each client data, we calculate the EMD distance on metadata available in training medical images from each client. This metadata includes maximum intensity values and label volume values related to specific use cases. For instance, in the Federation of Tumor Segmentation (FeTS), where three labels (WT: whole tumor, TC: tumor core, ET: expanding tumor) are segmented, we extract volume values corresponding to these labels. The choice of this metadata is based on the assumption that it can be consistently extracted for all experiments, making it inherently available for all medical radiological images.

Subsequently, the metadata vectors $V_{Mi}$ for $i = 1, 2, \ldots, N$ obtained from each client (where $N$ is the total number of federated clients), representing the extracted metadata, are transmitted to the server. Here, the EMD is computed for each client in relation to other clients within the federation. The choice of this distance is intentional as it has demonstrated its robustness to capture dissimilarity between probability distributions and thus provide valuable insights into the distributional disparities among clients (Luo et al., 2023). Other metrics may be considered in alternative domains.

## 2.2. Non-IID Assessment with Euclidean Embedding Distance

In contrast to computing non-IID solely based on metadata, which may not capture the intricate characteristics within the data, we investigate computation in the embedding space of data for its capacity to capture deeper and richer data representations. Given the domain (i.e., medical imaging), a large publicly-available pre-trained MedicalNet model (Chen et al.,

2019) is utilized for embedding extraction in our proposed methodology. Training on the diverse 3DSeg-8 dataset, covering a wide range of modalities, organs, and pathologies, has resulted in the development of a set of heterogeneous 3D neural network models. We adopt the pre-trained 3D-ResNet18 network to extract embedding features from the local data on each client. Specifically, image features are extracted from the network' bottleneck layer, known for providing concise and essential representations of input data. We compute the angles between Principal Component Analysis (PCA) components (where PCA = 2) to validate their alignment (more details in Appendix E). These components compress vectors before transmission to the server, thereby enhancing privacy as reconstruction accuracy drops by decreasing the number of components (Reddy and Jaya, 2021).

Subsequently, all client embedding vectors, denoted as $V_{Ei}$ for $i = 1, 2, \ldots, N$, each with a dimension of *client_samples*$\times$(512, 2) and flattened to *client_samples*$\times$(1024), are transmitted to the server. These vectors are then utilized for the computation of EUC according to Equation (1):

$$EUC(V_{Ei}, V_{Ej}) = \sqrt{\sum_{k=1}^{N}(V_{Eik} - V_{Ejk})^2},\tag{1}$$

where $V_{Ei}$ and $V_{Ej}$ represent feature vectors from any two clients.

### 2.3. Generalization and Personalization Strategies

For generalization, we propose downgrading the most distant client (outlier) in the federation. To achieve this, we utilize the distances between clients computed during the non-IID assessment step and incorporate this information into weights ($\omega$) assigned to each client. This approach is implemented using the FedAvg algorithm for the proof of concept, while noting that any FL averaging method could be employed. The proposed update of the global model in FedAvg$_w$ is defined in Equation (2):

$$w_{t+1} = w_t - \alpha \cdot \sum_{i=1}^{N} \omega_{i,t} \cdot \nabla f_i(w_t),\tag{2}$$

where $w_{t+1}$ is the updated global model, $w_{i,t}$ represents the weight for client $i$ at iteration $t$, $f_i(w_t)$ is the local objective function for client $i$, and $N$ is the total number of clients.

To identify a client for down-weighting, let $D_{\text{EMD}}$ and $D_{\text{EUC}}$ represent the matrices of EMD and EUC distances between clients, respectively. By summing along each column, we identify the client with the highest total (distant client) according to Equation (3), where $D$ represents either $D_{\text{EUC}}$ or $D_{\text{EMD}}$:

$$i_{max} = \arg\max_i \left(\sum_{j=1}^{N} D_{ij}\right).\tag{3}$$

Following that, we degrade its contribution by applying the arbitrary weight values ($\omega$) of 0.1, 0.3 and 0.5 to demonstrate the trend in improving the performance. This procedure enables us to assess the impact of reducing the influence of a single distant client ($i_{max}$) on the entire federation, thereby providing a clear illustration of the correlation between

computed distances and the performance of the global model. The base weights ($\omega$) for the non-downgraded clients are 1.

Then, in our personalization strategy, we use the same distance matrices $D_{\text{EMD}}$ and $D_{\text{EUC}}$ to build clusters of closely-related clients. The clustering algorithm minimizes total distances within each cluster (i.e., $C_1$ and $C_2$). The process is outlined in Algorithm (1).

---

**Algorithm 1:** Cluster Assignment

---

**Input:** Distance matrix $D$ and number of clients $N$
**Output:** Cluster assignments
// Step 1:  Identify the most distant client $i_{\text{max}}$
$i_{\text{max}} \leftarrow \arg\max_i \sum_j D_{ij}$;
$C_2 \leftarrow \{i_{\text{max}}\}$ ;                                    // Initialize cluster $C_2$ with $i_{\text{max}}$
$C_1 \leftarrow \{i \mid i \neq i_{\text{max}}\}$ ;              // Initialize remaining clients in cluster $C_1$
// Step 2:  Assign one or two closest clients $i_{\text{next}}$ to $i_{\text{max}}$
**while** $C_1$ *is not empty* **do**
  $i_{\text{next}} \leftarrow \arg\min_{i \in C_1} D_{i_{\text{max}}i}$;
  Assign $i_{\text{next}}$ to $C_2$;
  $C_1 \leftarrow C_1 \setminus \{i_{\text{next}}\}$ ;                                    // Remove $i_{\text{next}}$ from $C_1$
  **if** $C_1$ *has only 2 clients left* **then**
    Break ;                  // Break the loop if only two clients left in $C_1$
  **end**
**end**

---

This clustering approach promotes effective collaboration and information exchange among clients with closer data distributions within each cluster. We limited the evaluation to only two clusters to show the benefit of the clustering approach and its impact on improving performance per client, while maintaining a reasonable number of experiments.

## 3. Experiments

### 3.1. Datasets

We used three publicly available FL datasets for our study. FeTS 2021 (Pati et al., 2021) consists of glioblastoma multi-modality MRIs from multiple sites, with WT, TC, and ET segmentations. For our experiment, we selected four (4) clients, ensuring a balanced distribution of samples (Hospital$_6$: 34 samples, Hospital$_{13}$: 35 samples, Hospital$_{20}$: 33 samples, and Hospital$_{21}$: 35 samples). The multi-site prostate MRI segmentation dataset (Liu et al., 2020b), (Liu et al., 2020a) features T2-weighted MRIs with prostate segmentation masks. We used four (4) balanced clients for our experiments: Client$_1$: 39 samples, Client$_2$: 32 samples, Client$_3$: 40 samples, Client$_4$: 39 samples. The Fed-KITS2019 dataset (Terrail et al., 2022) focuses on kidney and tumor segmentation in CT scans. We created a 5-client federated version, excluding one site (Client$_6$: 30 samples) for a balanced distribution of samples: Client$_1$: 12 samples, Client$_2$: 14 samples, Client$_3$: 12 samples, Client$_4$: 12 samples, Client$_5$: 16 samples. Our focus on balanced and small federations promotes equal and rapid contributions from each client, facilitating equitable evaluation of the FL model

across datasets. The data in each group were divided into training (80%) and validation (20%) sets, as originally proposed.

Additionally, we redistributed labels among clients to create non-IID federations, diversifying client distributions. We followed the methodology proposed in (Luo et al., 2023). For non-IID federations, we aimed to maintain consistency in assigning training and validation samples across each set. However, if they were not available on the same client, adjustments were made, potentially resulting in differences in the selected samples, while preserving set sizes. Please refer to Appendix A for more details regarding the data, and Appendix D for information on building non-IID federations.

### 3.2. Training and Validation

We used a 3D U-Net network (Ronneberger et al., 2015) along with the SGD optimizer with a learning rate of 0.01 and momentum of 0.9, employing DiceLoss for training, following a standard protocol for medical segmentation. Training was conducted using FedAvg (McMahan et al., 2017), along with its weighted variant (Equation 2), across 25 global epochs as a balance between model convergence and optimized experiment time. We assessed performance using the Dice Metric, considering inter-client standard deviation for variation. We compared our methods against two representative FL algorithms: FedProx (Li et al., 2020) and DITTO (Li et al., 2021a). Please refer to Appendix B for further details.

## 4. Results

### 4.1. Generalization Optimization

To further reduce computation of $D_{EMD}$ for FeTS clients with many modalities and labels, we considered only the metadata where EMD values are maximum ($EMD_{max}$) among others as having the greatest negative impact. This applies to both maximum intensity ($EMD_{maxI}$) and label volume values ($EMD_{maxL}$). Please refer to Appendix C for more details. Table 1 presents the correlation between EMD and the performance of the FedAvg global model, indicating that as EMD values increase, performance decreases. We refine the federation with the *lowest performance*[A,B,C] to enhance generalization and personalization strategies.

Table 1: Comparison of $EMD_{maxI}$, $EMD_{maxL}$, and Dice score across different federations.

| Federations: | FeTS | | | | Prostate | | Fed-KITS2019 | |
|---|---|---|---|---|---|---|---|---|
| | Original | WT | TC | ET | Original | Prostate | Original | Kidney + Tumor |
| $EMD_{maxI}$ | 1.60 | 1.45 | 1.19 | 0.59 | 6.88 | 0.38 | 1.17 | 0.51 |
| $EMD_{maxL}$ | 1.56 | 6.31 | 6.83 | 10.96 | 0.48 | 9.83 | 0.87 | 11.62 |
| Dice | $0.867_{\pm0.10}$ | $0.856_{\pm0.04}$ | $0.843_{\pm0.05}$ | $0.828_{\pm0.12}$[A] | $0.325_{\pm0.10}$ | $0.271_{\pm0.06}$[B] | $0.459_{\pm0.03}$ | $0.423_{\pm0.02}$[C] |

Analyzing the distance matrices provides valuable insights into the relationships between different clients (Table 2). In FeTS[A], both EMD and EUC highlight $client_1$ as the most distant within the federation while for Prostate[B], $client_4$ emerges as the most distant. In Fed-KITS2019[C], while both metrics suggest significant differences between clients, the most distant client differs. This discrepancy is attributed to the nature of the data each distance operates on. EMD analyzes metadata distribution, while EUC operates in the embedding

Table 2: Distances matrices ($D_{EMD}$ and $D_{EUC}$) for FeTS[A], Prostate[B], and Fed-KITS2019[C].

| FeTS[A] Clients | | | | | | | | | |
|---|---|---|---|---|---|---|---|---|---|
| (EMD) | 1 | 2 | 3 | 4 | (EUC) | 1 | 2 | 3 | 4 |
| 1 | - | 3.60 | 7.87 | 13.91 | 1 | - | 23 | 49 | 63 |
| 2 | 3.60 | - | 1.75 | 4.55 | 2 | 23 | - | 30 | 47 |
| 3 | 7.87 | 1.75 | - | 2.98 | 3 | 49 | 30 | - | 22 |
| 4 | 13.91 | 4.55 | 2.98 | - | 4 | 63 | 47 | 22 | - |
| Sum | 25.38 | 6.30 | 4.73 | 21.45 | Sum | **135** | 100 | 101 | 132 |
| **Prostate[B] Clients** | | | | | | | | | |
| (EMD) | 1 | 2 | 3 | 4 | (EUC) | 1 | 2 | 3 | 4 |
| 1 | - | 1.24 | 3.01 | 8.54 | 1 | - | 51 | 83 | 122 |
| 2 | 1.24 | - | 2.70 | 11.05 | 2 | 51 | - | 60 | 98 |
| 3 | 3.01 | 2.70 | - | 4.31 | 3 | 83 | 60 | - | 59 |
| 4 | 8.54 | 11.05 | 4.31 | - | 4 | 122 | 98 | 59 | - |
| Sum | 12.79 | 13.75 | 7.02 | 23.90 | Sum | 256 | 209 | 202 | **279** |

| Fed-KITS2019[C] Clients | | | | | | | | | | | |
|---|---|---|---|---|---|---|---|---|---|---|---|
| (EMD) | 1 | 2 | 3 | 4 | 5 | (EUC) | 1 | 2 | 3 | 4 | 5 |
| 1 | - | 0.92 | 1.45 | 2.46 | 13.86 | 1 | - | 936 | 1268 | 2743 | 1207 |
| 2 | 0.92 | - | 1.03 | 2.37 | 12.55 | 2 | 936 | - | 844 | 2211 | 600 |
| 3 | 1.45 | 1.03 | - | 1.15 | 11.03 | 3 | 1268 | 844 | - | 1602 | 303 |
| 4 | 2.46 | 2.37 | 1.15 | - | 4.89 | 4 | 2743 | 2211 | 1602 | - | 1548 |
| 5 | 13.86 | 12.55 | 11.03 | 4.89 | - | 5 | 1207 | 600 | 303 | 1548 | - |
| Sum | 18.70 | 15.95 | 13.21 | 8.40 | 42.34 | Sum | 6154 | 4591 | 4017 | **8104** | 3658 |

space, potentially capturing different data features. As a result, certain clients may appear more distant in one assessment compared to the other.

Table 3 compares results across various learning approaches, including our down-weighting strategy (FedAvg$_w$) for distant clients. In FeTS[A], FedAvg$_w$ leads to a relative increase in performance compared to the default FedAvg approach and FedProx. For Prostate[B], it significantly enhances performance (+5% vs FedAvg, +16.6% vs FedProx). Similarly, in Fed-KITS2019[C], it improves Dice performance regardless of distance metric (EMD or EUC).

Table 3: Global model Dice scores (mean ± standard deviation between clients in the federation). EMD indicates Earth Mover's distance, EUC stands for Euclidean distance.

| Algorithm/Dataset: | FeTS[A]: Client$_1$ | Prostate[B]: Client$_4$ | Fed-KITS2019[C] EMD: Client$_5$ | Fed-KITS2019[C] EUC: Client$_4$ |
|---|---|---|---|---|
| FedAvg, $\omega_{default} = 1.0$ | $0.828_{\pm 0.12}$ | $0.271_{\pm 0.06}$ | $0.423_{\pm 0.02}$ | $0.423_{\pm 0.02}$ |
| FedProx, μ = 0.1 | $0.831_{\pm 0.13}$ | $0.155_{\pm 0.07}$ | $0.395_{\pm 0.02}$ | $0.395_{\pm 0.02}$ |
| FedAvg$_w$, $\omega = 0.1$ | $0.812_{\pm 0.14}$ | $\mathbf{0.321}_{\pm 0.06}$ | $\mathbf{0.438}_{\pm 0.02}$ | $0.449_{\pm 0.02}$ |
| FedAvg$_w$, $\omega = 0.3$ | $0.833_{\pm 0.14}$ | $0.255_{\pm 0.06}$ | $0.428_{\pm 0.02}$ | $\mathbf{0.460}_{\pm 0.04}$ |
| FedAvg$_w$, $\omega = 0.5$ | $\mathbf{0.840}_{\pm 0.12}$ | $0.290_{\pm 0.07}$ | $0.409_{\pm 0.03}$ | $0.456_{\pm 0.03}$ |

## 4.2. Personalization Optimization

According to Table 2, for FeTS[A] and Prostate[B] clients, two distinct clusters based on minimum distances could be formed, facilitating potential collaboration and improving model

performance within FL. Fed-KITS2019$^C$ clients also exhibit clustering, with differing perspectives from EMD and EUC. While EMD provides only one clear cluster assignment, with EUC, we explore another assignment option based on the proximity of client$_3$ to client$_5$ as well.

Table 4: Dice scores for FeTS$^A$, Prostate$^B$, Fed-KITS2019$^C$ personalization optimization.

| Algorithm/Clients: | 1 | 2 | 3 | 4 | 5 | Average |
|---|---|---|---|---|---|---|
| FeTS$^A$ | | | | | | |
| FedAvg$_{default}$ | $0.662_{\pm0.28}$ | $0.824_{\pm0.10}$ | $0.897_{\pm0.11}$ | $0.929_{\pm0.04}$ | - | 0.828 |
| DITTO | $0.698_{\pm0.25}$ | $0.832_{\pm0.08}$ | $0.893_{\pm0.10}$ | $0.938_{\pm0.03}$ | - | 0.840 |
| FedAvg$_{\{1,2\}\{3,4\}}$ | $\mathbf{0.702}_{\pm0.25}$ | $\mathbf{0.864}_{\pm0.06}$ | $\mathbf{0.914}_{\pm0.07}$ | $\mathbf{0.947}_{\pm0.03}$ | - | **0.857** |
| Prostate$^B$ | | | | | | |
| FedAvg$_{default}$ | $0.194_{\pm0.11}$ | $0.245_{\pm0.13}$ | $0.316_{\pm0.14}$ | $0.330_{\pm0.09}$ | - | 0.271 |
| DITTO | $0.232_{\pm0.12}$ | $0.276_{\pm0.14}$ | $\mathbf{0.343}_{\pm0.13}$ | $0.355_{\pm0.10}$ | - | 0.302 |
| FedAvg$_{\{1,2\}\{3,4\}}$ | $\mathbf{0.337}_{\pm0.17}$ | $\mathbf{0.347}_{\pm0.15}$ | $0.337_{\pm0.11}$ | $\mathbf{0.422}_{\pm0.12}$ | - | **0.361** |
| Fed-KITS2019$^C$ | | | | | | |
| FedAvg$_{default}$ | $0.398_{\pm0.36}$ | $0.4338_{\pm0.37}$ | $\mathbf{0.439}_{\pm0.35}$ | $0.414_{\pm0.39}$ | $0.428_{\pm0.25}$ | 0.423 |
| DITTO | $0.367_{\pm0.33}$ | $0.429_{\pm0.35}$ | $0.400_{\pm0.32}$ | $0.400_{\pm0.39}$ | $0.430_{\pm0.23}$ | 0.405 |
| FedAvg$_{EMD:\{1,2,3\}\{4,5\}}$ | $0.441_{\pm0.37}$ | $0.437_{\pm0.38}$ | $0.433_{\pm0.39}$ | $0.468_{\pm0.38}$ | $\mathbf{0.594}_{\pm0.22}$ | **0.475** |
| FedAvg$_{EUC:\{1,2\}\{3,4,5\}}$ | $\mathbf{0.442}_{\pm0.39}$ | $\mathbf{0.446}_{\pm0.39}$ | $0.433_{\pm0.40}$ | $\mathbf{0.497}_{\pm0.41}$ | $0.528_{\pm0.30}$ | 0.469 |

Table 4 compares Dice scores for personalized models across different clients, revealing insights into method performance. In FeTS$^A$, FedAvg$_{\{1,2\}\{3,4\}}$ consistently outperforms FedAvg$_{default}$ and DITTO, indicating improved segmentation with personalized learning based on $C_1 = \{1,2\}$ and $C_2 = \{3,4\}$. Similarly, in Prostate$^B$, FedAvg$_{\{1,2\}\{3,4\}}$ shows significant improvements over FedAvg$_{default}$ and DITTO. In Fed-KITS2019$^C$, segmentation either through EMD or EUC clustering consistently outperforms FedAvg$_{default}$ and DITTO.

## 5. Conclusion and Discussion

Our study underscores the significance of assessing client data heterogeneity (non-IID) in medical imaging FL to optimize both generalization and personalization goals. We propose a down-weighting strategy to enhance global model performance across datasets by reducing the impact of a distant client. Additionally, we advocate for distance-based clustering of clients as a personalization solution to enhance medical imaging segmentation accuracy across diverse datasets.

While promising, our study is limited to medical imaging, particularly volumetric radiographic datasets, and prioritizes balanced scenarios and small federations for faster computation and proof-of-concept purposes. Future research should explore unbalanced scenarios, larger federations, alternative architectures for embedding extraction to broaden the applicability of our proposed strategies.

## Acknowledgments

This work was partially supported by the Region Grand Est (project CLINNOVA) and by French State Funds managed by the Agence Nationale de la Recherche (ANR) under Grant ANR-22-FAI1-0001 (project DAIOR) and Grant ANR-10-IAHU-02 (IHU Strasbourg).

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

## Appendix A. Detailed Data Information

Table 5 shows the number of samples in the training and validations splits of three segmentation datasets used in this study.

Table 5: Number of samples in the training and validation splits of three datasets.

|  | FeTS | | | | Prostate | | | | Fed-KITS2019 | | | | |
|---|---|---|---|---|---|---|---|---|---|---|---|---|---|
| Client: | 1 | 2 | 3 | 4 | 1 | 2 | 3 | 4 | 1 | 2 | 3 | 4 | 5 |
| Training | 34 | 35 | 33 | 35 | 39 | 32 | 40 | 39 | 9 | 11 | 9 | 9 | 12 |
| Validation | 7 | 7 | 7 | 7 | 7 | 6 | 8 | 7 | 3 | 3 | 3 | 3 | 4 |

We standardized our pre-processing and augmentation pipelines across all datasets to uphold consistency and reduce their potential impact on results. While maintaining uniform practices like ensuring channel-first representation and intensity normalization, we adjusted specific parameters, such as spacing and cropping sizes for four patch extraction (number of patches = 4) during training, to match the unique characteristics of each dataset.

For example, in the FeTs dataset, we utilized a spacing of (1.0, 1.0, 1.0), and a cropping size of (224, 224, 144). In the Prostate dataset, the spacing was set to (0.3, 0.3, 1.0), and the cropping size to (224, 224, 32). Meanwhile, for the Fed-KITS2019 dataset, we employed a spacing of (2.90, 1.45, 1.45), and a cropping size of (256, 256, 64).

As for augmentation, random flipping is applied along each spatial axis with a probability of 50%. Intensity scaling and shifting are applied with factors and offsets of 0.1, respectively, with a probability of 100%.

## Appendix B. Training and Validation

For the FeTS dataset, we utilized a batch size of 1, and sliding window inference with a window size of (240, 240, 160) was applied. Similarly, for the Prostate dataset, the batch size remained at 1, and sliding window inference was conducted with a window size of (224, 224, 32). For the Fed-KITS2019 dataset, we maintained a batch size of 1, and sliding window inference was performed with a window size of (256, 256, 80).

All training and validation processes were conducted using the MONAI [1] and NVIDIA FLARE [2] frameworks.

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

Table 6: $D_{EMD}$ matrix for the FLAIR modality and the ET (Enhancing Tumor) label.

| | FLAIR | | | | Enhancing Tumor (ET) | | | |
|---|---|---|---|---|---|---|---|---|
| $(\text{EMD}_{maxI})$ | 1 | 2 | 3 | 4 | $(\text{EMD}_{maxL})$ | 1 | 2 | 3 | 4 |
| 1 | - | 0.33 | 0.32 | 0.27 | 1 | - | 6.87 | 15.42 | 27.55 |
| 2 | 0.33 | - | 0.34 | 1.45 | 2 | 6.87 | - | 3.07 | 7.40 |
| 3 | 0.32 | 0.34 | - | 0.86 | 3 | 15.42 | 3.16 | - | 5.18 |
| 4 | 0.27 | 1.45 | 0.86 | - | 4 | 27.55 | 7.40 | 5.18 | - |

## Appendix C. Computation of FeTS Distance Matrix

As an example, we calculate the final $D_{EMD}$ matrix for the non-IID ET federation of FeTs$^A$ clients. Initially, we compute the Earth Mover's Distance (EMD) for intensities of all modalities across all clients: T2-weighted, T1-weighted, T1-weighted with contrast enhancement (T1C+), and FLAIR. This provides us with the following values: $\text{EMD}_{T2} = 0.393$, $\text{EMD}_{T1} = 0.247$, $\text{EMD}_{T1C+} = 0.290$, and $\text{EMD}_{FLAIR} = 0.595$.

Similarly, we perform the same process for the three available labels for segmentation: WT (Whole Tumor), TC (Tumor Core), and ET (Enhancing Tumor) for each client with respect to each other client. We obtain the following EMD values: $\text{EMD}_{WT} = 0.935$, $\text{EMD}_{TC} = 0.561$, and $\text{EMD}_{ET} = 10.96$.

We select the maximum EMD in intensity modalities as $\text{EMD}_{maxI} = 0.595$, corresponding to the FLAIR modality ($\text{EMD}_{FLAIR}$), and the maximum EMD in label distributions as $\text{EMD}_{maxL} = 10.96$, corresponding to the ET (Enhancing Tumor) label ($\text{EMD}_{ET}$). Consequently, we construct two $D_{EMD}$ matrices representing client-to-client correlation based on these values, as presented in Table 6.

Table 7: Final $D_{EMD}$ matrix for the FeTS$^A$ clients.

| $D_{EMD}$ for FeTS$^A$ | | | |
|---|---|---|---|
| | 1 | 2 | 3 | 4 |
| 1 | - | 3.60 | 7.87 | 13.91 |
| 2 | 3.60 | - | 1.70 | 4.43 |
| 3 | 7.87 | 1.75 | - | 3.02 |
| 4 | 13.91 | 4.43 | 3.02 | - |

For the final $D_{EMD}$ matrix, we average these two matrices, with the result shown in Table 7.

## Appendix D. Building Non-IID Federations

As an example of building non-IID federations, we outline the main steps. In accordance with (Luo et al., 2023), we utilized label sizes as a criterion for forming such federations. We redistributed label sizes by organizing them from smallest to largest. Specifically, for the FeTS dataset, this process was conducted for each specific use case (WT, TC, ET); for the Prostate dataset, it was done for the prostate label; and for Fed-KITS2019, we examined both kidney and tumor regions collectively.

Figures 2, 3, and 4 represent distributions of labels for each non-IID federation.

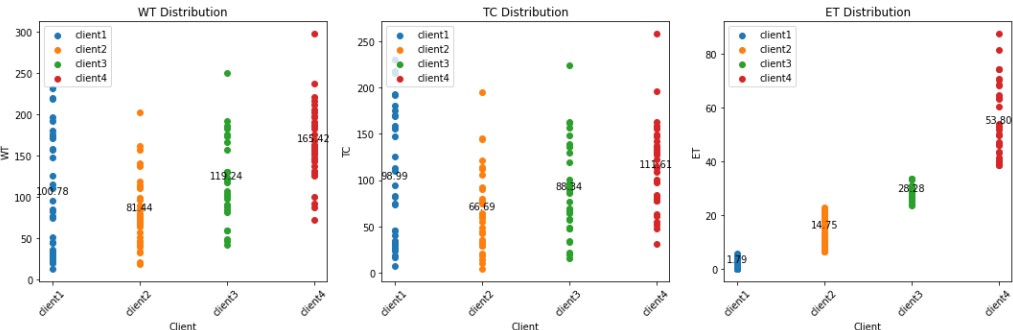

Figure 2: Distribution of tumor labels (WT, TC, ET) for FeTS[A]: ET non-IID federation.

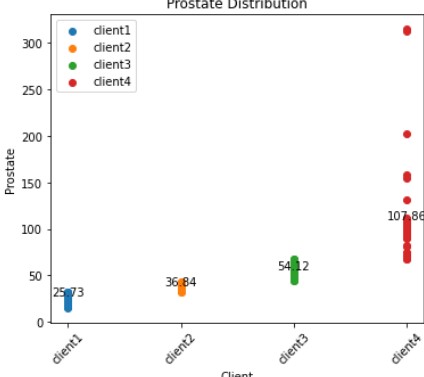

Figure 3: Distribution of the prostate label for Prostate[B]: Prostate non-IID federation.

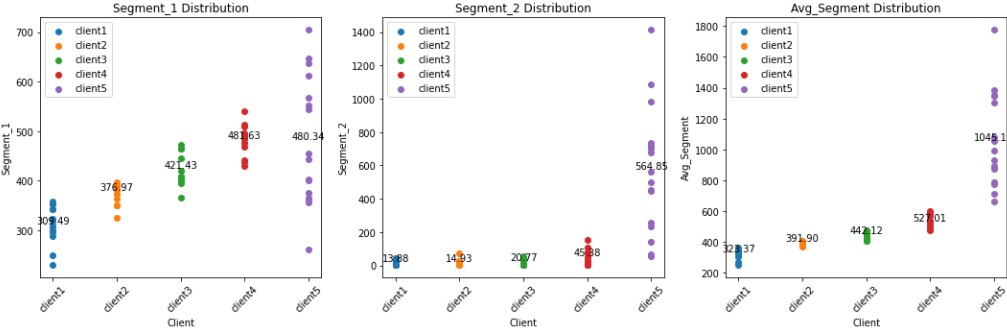

Figure 4: Distribution of Segment_1 (kidney), Segment_2 (tumor), and average (Avg) labels for Fed-KITS2019[C]: Kidney+Tumor non-IID federation.

## Appendix E. Angles of PCA Components

Figures 5, 6, and 7 show the angles between PCA components (1 and 2) for each non-IID federation.

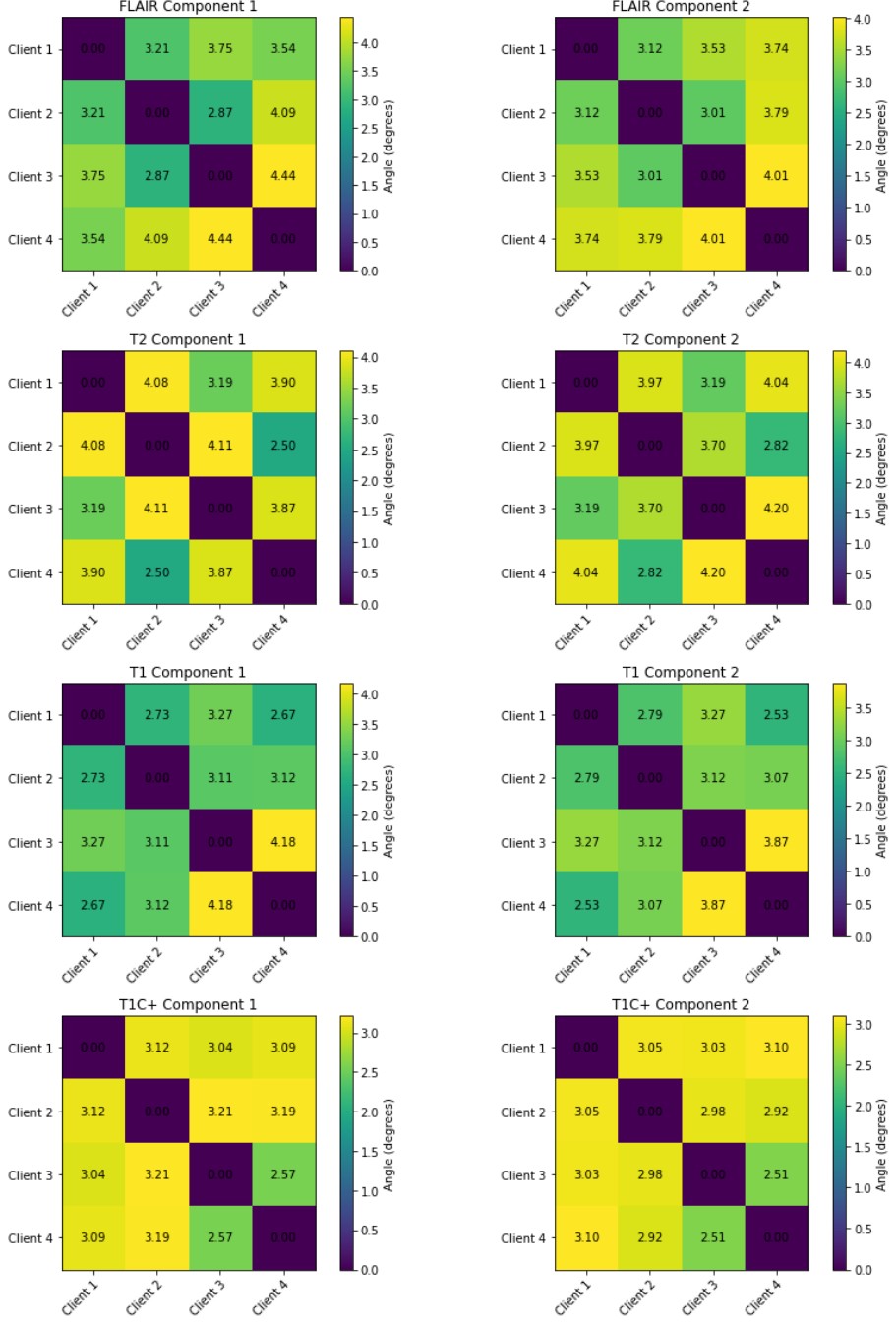

Figure 5: Angles between PCA components (1 and 2) for the FeTS$^A$ clients.

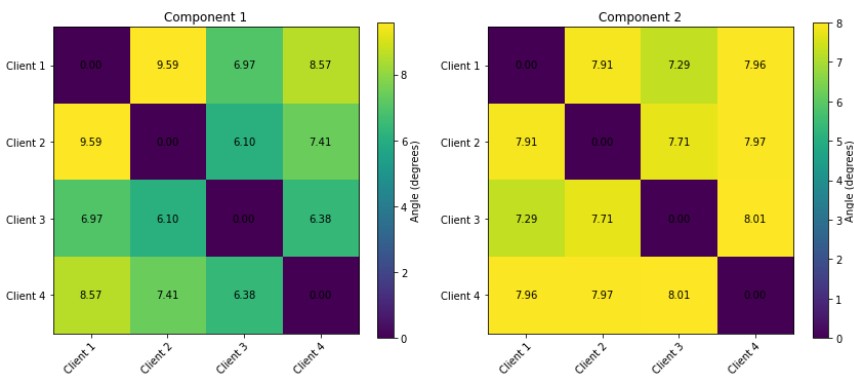

Figure 6: Angles between PCA components (1 and 2) for the Prostate$^B$ clients.

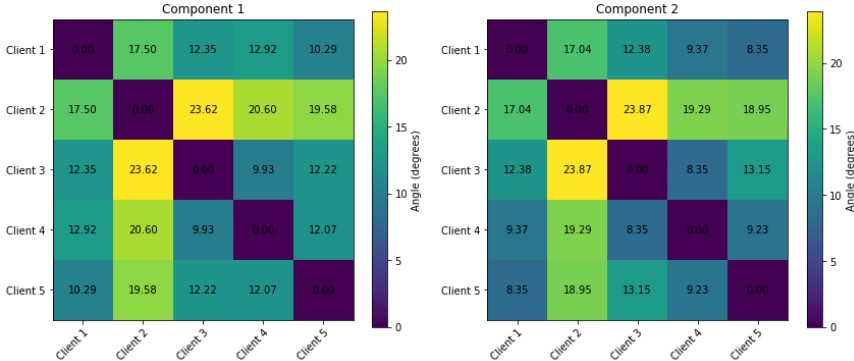

Figure 7: Angles between PCA components (1 and 2) for the Fed-KITS2019$^C$ clients.

The formula to compute the angle between two PCA components $\mathbf{v}$ and $\mathbf{w}$ is given by:

$$\text{Angle}(\mathbf{v}, \mathbf{w}) = \arccos\left(\frac{\mathbf{v} \cdot \mathbf{w}}{\|\mathbf{v}\| \cdot \|\mathbf{w}\|}\right)$$

where $\cdot$ represents the dot product, $\|\mathbf{v}\|$ and $\|\mathbf{w}\|$ represent the magnitudes of vectors $\mathbf{v}$ and $\mathbf{w}$ respectively, and arccos is the inverse cosine function. We iterate over pairs of clients and computes the angle between the corresponding components for a random sample on each client.

