# OpenReview forum: "Distance-Aware Non-IID Federated Learning for Generalization and Personalization in Medical Imaging Segmentation"
_MIDL.io/2024/Conference — MIDL 2024 Poster_

### Official Review · Reviewer_aSJy · 2024-02-16

**Confidence:** 5
**Preliminary Rating:** 4

**Summary:**

This work presents a practical non-IID assessment methodology for a medical segmentation problem, highlighting its significance in medical FL.

**Strengths:**

This work presents a practical non-IID assessment methodology for a medical segmentation problem, highlighting its significance in medical FL. The result looks good. However, I still have some concerns:

**Weaknesses:**

The first concern is regarding the novelty. The non-iid is a common issus in federated learning, as well as MRI segmentation. And the authir use a very commonly used technique to solve it, i.e., personalized FL. For example, Specificity-Preserving Federated Learning for MR Image Reconstruction TMI 2022. It is hard to see the novelty of this work. Thus, I would like to see the motivation for the author to use such a method or the new contribution of this work, not only the downstream task, i.e., segmentation.

The second concern is regarding the related works, I would like to see a more complete review on the medical FL. Actually, benefited from the large scale pretrained models, many medical FL also build on them to significantly improve the medical FL performance. For example, Learning Federated Visual Prompt in Null Space for MRI Reconstruction, CVPR 2023.

**Detailed Comments:**

see the Weaknesses

**Justification Of The Preliminary Rating:**

I would like to see the author add more comparisons with other related works that I have mentioned. The novelty and contribution also need to be clarified, because the current version lacks strong motivation.

**Questions To Address In The Rebuttal:**

see the Weaknesses

---

> ### Author Response · Authors · 2024-03-15
> **Reviewer Recommendation for Citation**
>
> Dear Program and Area Chairs,
>
> We have carefully considered all the comments and suggestions provided by the reviewer. However, I would like to bring to your attention a specific request made by Reviewer aSJy.
>
> Reviewer aSJy has expressed satisfaction with our manuscript overall but has recommended the inclusion of citations to two papers, namely:
>
> 1. Feng, Chun-Mei, et al. "Specificity-preserving federated learning for MR image reconstruction." IEEE Transactions on Medical Imaging (2022).
> 2. Feng, Chun-Mei, et al. "Learning federated visual prompt in null space for MRI reconstruction." Proceedings of the IEEE/CVF Conference on Computer Vision and Pattern Recognition. 2023.
>
> We are more than willing to include any citations that directly contribute to the manuscript's scientific merit. However, both of these papers focus on MRI reconstruction, whereas ours explores medical image segmentation. These papers are not directly related to our paper's scope, methodology, or findings. As it is evident that these papers are from the same author, we believe Reviewer aSJy has asked us to cite these papers to improve their citations. It is necessary to bring to your attention that such a practice should be discouraged. Furthermore, incorporating these unrelated citations might detract from the coherence of our paper, potentially confusing readers regarding the focus and contribution of our study.
>
> Therefore, we kindly request your guidance on proceeding with this recommendation while maintaining the submission's integrity.
>
> Thank you!

---

> > ### Comment · Area_Chair_YH9e · 2024-03-15
> > **no need to cite if not relevant**
> >
> > Thank you for bringing this up.
> >
> > There is no need to cite if you do not think that these papers are relevant, you can address the reviewers comments by including citations to other more relevant work in your opinion. This has been noted and will be kept in mind during decisions.
> >
> > Best regards and thanks for engaging with the review process!

---

> ### Author Response · Authors · 2024-03-15
>
> Dear Reviewer,
>
> Thank you for your thorough evaluation of our work and for providing detailed feedback on the novelty and contribution of our methodology. We have carefully considered your comments and would like to address each concern raised.
>
> Regarding the novelty and contribution of our methodology, we understand your perspective on the familiarity of non-IID issues and personalized federated learning techniques within the context of medical federated learning (FL). However, we believe our work offers several significant contributions to the field.
>
> Firstly, our methodology introduces a practical non-IID assessment specifically tailored for medical segmentation problems. Our work is related to the area of personalization, but we are broadly examining the question of non-IID in medical FL. In particular, we propose a methodology that not only effectively measures non-IID but also serves as a tool for improving performance within federation. We find this practically useful because working with a new dataset requires initial steps to measure the level of non-IID in order to choose the optimal strategy. The decision to consider the segmentation use case is based on its interest from the community and the number of available federated learning benchmarks (FeTS, FLamby, FL-PoST, and others), which provide the basis for fair comparison. Additionally, we observe that there has not been much research done on medical image segmentation and non-IID, as also stated by Luo G, Liu T, Lu J, Chen X, Yu L, Wu J, Chen DZ, Cai W. Influence of Data Distribution on Federated Learning Performance in Tumor Segmentation. Radiol Artif Intell. 2023 Apr.
>
> We appreciate your reference to "Specificity-Preserving Federated Learning for MR Image Reconstruction TMI 2022" as a relevant work in the field of personalized strategies. Indeed, the approach proposed in that work aligns with our literature review. The authors propose to divide the MR reconstruction model into two parts: a globally shared encoder to obtain a generalized representation at the global level, and a client-specific decoder to preserve the domain-specific properties of each client. In the reference (Krishna Pillutla, Kshitiz Malik, Abdel-Rahman Mohamed, Mike Rabbat, Maziar Sanjabi, and Lin Xiao. Federated learning with partial model personalization) the authors consider this way of partitioning deep learning models where some layers (like decoder part can be personalized), which is exactly proposed in Specificity-Preserving Federated Learning for MR Image Reconstruction TMI 2022. To sum up, while the area of application differs, with our focus on medical segmentation rather than MRI reconstruction, the fundamental concept of “partitioning” deep learning models to enable personalization remains consistent.
>
> We believe that the work you highlighted “Learning Federated Visual Prompt in Null Space for MRI Reconstruction”, as well as, for example, "FedFMS: Exploring Federated Foundation Models for Medical Image Segmentation," align with our research direction proposed in the manuscript. While the use of large-scale pretrained models is indeed a promising avenue for improving FL performance, we believe that it could be also referred to the pretraining methods (the authors again use the ideas of pretraining on massive data and later fine-tuning). Our focus is on the importance of measuring non-IID data and strategically splitting clients into groups based on distance measurements or using degradation of a distant client. In fact, our methodology could potentially enhance the effectiveness of pretrained models by ensuring that clients with similar data distributions are grouped together for more efficiency.
>
> Thank you for bringing this to our attention as possible further direction of our research!

---

### Official Review · Reviewer_c6rg · 2024-02-27

**Confidence:** 3
**Preliminary Rating:** 3
**Final Rating:** 3.5

**Summary:**

This work develops novel federated learning techniques to mitigate the loss in performance due to the non-IID nature of realistic decentralized datasets. The authors propose two distances to quantify the degree of dissimilarity of the data distribution of two clients in a federated fashion. They use these distances both in a rescaling fashion in standard weighted Federated Averaging (FedAvg), and as a distance matrix in standard clustered federated learning. The method is validated on three different medical image segmentation datasets.

**Strengths:**

- The authors propose interesting non-IID metrics computable in a federated fashion for 3D segmentation tasks on MRI and CT modalities, of particular interest in federated medical image analysis,
- They explore the usage of these metrics in two known federated frameworks, e.g. re-weighted FedAvg and clustered FedAvg.
- Figure 1 greatly summarizes the proposed methods,
- Experiments on interesting well balanced subsets of multiple segmentation datasets were led.

**Weaknesses:**

**Euclidean Embedding Distance computation** The description of the computation of this distance is unclear to me. In particular page 4 on client embedding vectors
> "Principal Component Analysis (PCA) with 32 components is used to compress them before transmission to the server, enhancing privacy",

followed by

> "Subsequently, all client embedding vectors, denoted as $V_{Ei}$ for i = 1, 2, . . . , N , each with a dimension of (512, 32) and flattened to (16384), are transmitted to the server."

I am unsure where does the dimension (512,32) for the vector of one client come from. Moreover, how is the PCA actually computed? I would assume either on server-side after transmission, or in a federated fashion? It seems to me that the transmitted vectors $V_{Ei}$ and $V_{Ej}$ of two clients which independently computed a PCA do not lie in the same sub-space, I don't think an euclidean distance between them would be usable.


**Clustering algorithm** The choice of clustering algorithm seems quite arbitrary and task specific. It always gives a cluster $C_1$ of 2 clients. I understand that small federations is the use case of interest, but more established clustering algorithms such as spectral or agglomerative clustering would potentially make the method more broadly applicable. Could you please elaborate on the choice of the clustering algorithm?

**Experiments description** The experiments of part 4.1 are only partially described. I could not find information on the actual non-IID splits apart from a citation, even in Appendices. It makes reading Table 1 difficult and further results quite misleading. It would greatly improve the readability of the paper to add more information on the different federations, potentially in Appendix, as well as writing the federation used for each dataset in Tables 2, 3 and 4 in the captions as I am still unsure about this information.

**Comparison to other methods** Only few state-of-the-art methods are compared in the paper. I don't think it is a problem for part 4.1 as its goal as I understand is mainly to validate the usage of the introduced non-IID distances. However, the clustering federated learning results would benefit to be compared with state of the art clustered methods, with the closest to my knowledge being CFL : Avishek Ghosh et al., “An Efﬁcient Framework for Clustered Federated Learning,”. The only difference with the proposed method lies in the distance matrix as they use the cosine similarity between local updates of two clients. If possible, comparing the clustering results of CFL and the proposed method could strengthen the results.

**Detailed Comments:**

- Table 6: The computation of the $D_{EMD}$ final matrix for FeTS implies averaging distances with very different scales. It seems like the label information is the only one actually used in the final matrix. Did you try other ways of aggregating these matrices to avoid that, is it desired?
- Figure 1: for clarity, it would be better to name the last sample of a client with $Sample_{n_i}$ in the image on the right instead of $Sample_{N}$, $N$ is used for the number of clients.
- Equation 2: it should be written in the Method part that the base weights $w_{i,t}$ for the non-downgraded clients are 1 (if I understood well). Usually these weights are the proportion of the samples for a client $\rho_i = \frac{N_i}{N_{tot}}$, making the downgrading weights from 0.5 to 0.1 hard to understand.

**Justification Of Final Rating:**

I appreciated the discussion period with the authors. The clarification of the results as well as the addition of different Appendices greatly improved the readability of the paper and its value. I think that both the two proposed Non-IID metrics and the choice of datasets to work on are interesting and the community would benefit from it being published.

However, some of my concerns remain. The choice of clustering algorithm is questionable, although researchers applying this method could use something else. For one of the Non-IID metrics (EUC distance), clients compute local PCAs before communicating their embeddings and computing euclidean distances between these server-side. Appendix E shows that the components of each local PCA can be quite different from one client to another, confirming my concern. The angles between principal components of different clients are relatively small (at least for the first two components) for datasets FeTS and Prostate, but quite high for the last dataset. I cannot say if one can fully trust the current results for the EUC distance matrices for this last dataset, although in practice this problem could be handled by Federated PCAs or local PCA components communications, making the method probably viable.

Thus I keep my rating to borderline. I could consider it accepted depending on MIDL standards, if the authors can add in the article one or two sentences to explain why Appendix E is added, and what can be done to counter this problem.

**Justification Of The Preliminary Rating:**

The strengths of the paper would make me rate it as a 4: Weak accept at least. However, I think that the paper is too unclear for now on the description both of the method, especially the exact computation of the Euclidean Embedding Distance which might hide some problems, and of the experiments. I will definitely consider improving the rating when these points are clarified.

**Questions To Address In The Rebuttal:**

See the weaknesses above
- Clarification of the computation of the EUD,
- Clarification of the experiments,
- Some comments on the clustering algorithm,
- Ideally a comparison with CFL.

---

> ### Author Response · Authors · 2024-03-15
>
> Dear Reviewer,
>
> Thank you for your thorough review of our manuscript, and for providing detailed feedback on both the strengths and weaknesses of our work. We appreciate the time and effort you have invested in evaluating our research. We have carefully considered each of your comments and would like to address them below:
>
> Euclidean Embedding Distance computation: To clarify, the dimension (512,32) for the vector of one client is derived from the embedding vectors obtained from the client's local data, specifically from one MRI or CT scan (producing (512, 32)). We applied PCA=32 at the end of the bottleneck layer of the pre-trained network to compress these vectors. In our manuscript revision, we will provide a more detailed explanation of this computation. Regarding the question of whether computing PCA in this separate manner could affect the value of such spaces for Euclidean Distance computations, we believe that two key factors support its validity: 1) the consistent use of the same preprocessing data pipeline, and 2) the utilization of the same pretrained feature extractor (MedicalNet). These two points lead us to assume that the final features reside in the same subspace.
>
> Clustering algorithm: We acknowledge your concern regarding the choice of clustering algorithm and its potential impact on the method's applicability. While the current clustering algorithm may seem arbitrary and task-specific, we intentionally opted for simplicity and suitability for small federations, aligning with our specific use case. Our goal was to demonstrate that even with a straightforward clustering method, we can effectively enhance performance using our proposed methodology. The value of our approach lies not in the sophistication of the clustering algorithm, but rather in how we implement its clustering and what distance we rely on. Furthermore, it is important to note that our methodology is not limited to a particular clustering approach. Any clustering method aimed at grouping closely-related clients could be integrated with our proposed technique.
>
> Experiments description: We apologize for the lack of detailed description of the experiments in Section 4.1. In our Datasets Section (3.1), we briefly outlined the process of obtaining the federation by redistributing labels among clients to create non-IID federations, diversifying client distributions. This methodology was based on the approach proposed in Luo G, Liu T, Lu J, Chen X, Yu L, Wu J, Chen DZ, Cai W. Influence of Data Distribution on Federated Learning Performance in Tumor Segmentation. Radiol Artif Intell. 2023 Apr, specifically focusing on label size as a criterion. To ensure consistency in assigning training and validation samples across each set within non-IID federations, we aimed to maintain uniformity. However, adjustments were made if the required samples were not available on the same client, potentially resulting in differences in the selected samples while preserving set sizes. Though, we will provide additional information on the non-IID federations used in the Appendix to clarify the experimental setup.
>
> The other minor improvements proposed by you will be integrated for better readability.
>
> Thank you once again for your valuable feedback!

---

> > ### Comment · Reviewer_c6rg · 2024-03-20
> > **Insufficient justification of client local PCAs**
> >
> > Dear Authors,
> >
> > I first thank you for your rebuttal. Most of my remarks were relatively minor apart from my concern about client local PCAs. I agree with what is said for these minor comments, and greatly appreciated the modifications of the article, especially Appendix D.
> >
> > However, I don't think that your argument on the validity of computation of EUC is enough. Clients do perform independent PCAs in your method, there is a possibility that the embedding vectors transmitted to the server are not comparable using a simple coordinate-based euclidean distance. I understand that the same model and pre-processing were used to obtain the pre-PCA embeddings. But at the same time, this model is used to capture differences between the probably heterogeneous data of different clients. I would not be surprised if their computed PCAs were at least slightly different. This "slightly" is the important part to me, we just do not want the differences in PCAs to affect to much the computed distances. I think this could be solved by either a client computing its PCA and communicating it to the others, using Federated PCAs (https://arxiv.org/pdf/1907.08059.pdf), or something else anyway.
> >
> > Thus, I do not question the method in itself which I find very interesting, just the current definition of the PCAs and the presented values in the EUC distance matrices of the paper *which, without further verification, could be random*. For the sake of validation of the paper experiments, I would please ask you to compute angles between the principal directions of each client (angles between first directions of client 1, ..., 5, angles between second directions of 1, ..., 5, etc.) or any other form of distance between projection subspaces for at least one or two datasets, to verify your argument for the experiments of the paper. A small discussion of this aspect could be added in the Method part, proposing either local PCAs or another form for future applications of your method.
> >
> > All the best,
> > -.

---

> > > ### Author Response · Authors · 2024-03-25
> > >
> > > Dear Reviewer,
> > >
> > > Thank you for your active participation in the discussion and review of our manuscript. We appreciate your suggestion to compute angles between the principal directions of each client.
> > >
> > > In response to your concern, we have taken steps to compute the angles between components and have incorporated the findings into our manuscript, as detailed in Appendix E. We adjusted the step of the PCA computation by choosing the components with the smallest angles between them (in particular, 2 components). Additionally, we recomputed the distances in Table 2. We obtained the same findings in terms of both a distant client and clustering assessment.
> > >
> > > We believe that these revisions enhance the clarity of the computation of PCA on the client side. However, we acknowledge that Federated PCA could be considered for further studies, as proposed.
> > >
> > > Thank you once again for your feedback!

---

### Official Review · Reviewer_RSgw · 2024-02-28

**Confidence:** 4
**Preliminary Rating:** 3
**Final Rating:** 3.5

**Summary:**

This paper addresses non-IID data challenges in healthcare federated learning (FL) by proposing a novel assessment methodology for medical segmentation, leveraging distance measurements and metadata analysis to enhance FL performance. It introduces a novel approach in medical imaging FL that minimizes outlier effects and employs distance-based clustering for model personalization. Validated on three radiology imaging datasets, the proposed method significantly improves model generalization and personalization across diverse scenarios.

**Strengths:**

- This paper is well-motivated and the proposed framework is simple and easy to understand.
- The paper introduces a novel method that effectively addresses non-IID data challenges in medical imaging federated learning by utilizing Earth Mover’s Distance (EMD), significantly enhancing model generalization and personalization.
- The experiments are comprehensive, including several key baselines, generalization and personalization performance comparisons.

**Weaknesses:**

- The experiments fail to highlight the key hyperparameters influencing the proposed method's performance and omit some ablation studies on these significant hyperparameters.
- Algorithm 1 appears to be not self-contained and need more captions; the variable "i" is unclear, as it is also used as an index for D_{i,j}, creating confusion.

**Detailed Comments:**

N/A

**Justification Of Final Rating:**

I am satisfied with the response regarding concerns about hyper-parameters and have increased my score to a borderline accept. It would be beneficial to emphasize the key hyper-parameters in the experimental section. Additionally, using 'i' to represent individual clients seems unnatural, as it is commonly used to denote an index.

**Justification Of The Preliminary Rating:**

The evaluations of generalization and personalization are thorough overall, yet the paper misses crucial ablation studies. Additionally, certain algorithm descriptions require further elaboration and detailed clarification, particularly concerning key hyperparameters that might influence performance.

**Questions To Address In The Rebuttal:**

N/A

---

> ### Author Response · Authors · 2024-03-15
>
> Dear Reviewer,
>
> Thank you for the detailed review of our submission. We appreciate the time and effort you have put into evaluating our work. We would like to address the concerns raised regarding hyperparameters and algorithm clarity.
>
> Hyperparameters: We acknowledge your feedback on hyperparameters. However, it is important to note that our proposed method intentionally aims for simplicity and general applicability. The algorithm primarily leverages classical mathematical tools and existing metrics, such as Earth Mover’s Distance and Euclidean Distance, rather than introducing additional hyperparameters. The only new hyperparameter introduced in our approach is the weight parameter mentioned, particularly in Table 3, which adjusts the contribution of a distant client to the overall model generalization process. We provided the values for this parameter in the article to ensure transparency and reproducibility. Furthermore, the algorithm used for clustering, as detailed in the manuscript, does not depend on hyperparameters. It employs distance-based clustering techniques that are well-established and do not require tuning specific parameters for effective implementation. In summary, while we appreciate the importance of hyperparameters in machine learning algorithms, our approach intentionally minimizes their presence to maintain simplicity and applicability across different scenarios.
>
> Algorithm Clarity: To address the weakness regarding the clarity of Algorithm 1 and the confusion surrounding the variable "i," we would like to provide additional clarification. In the manuscript, when presenting the generalization and personalization strategies, we clarify that the variable "i" represents individual clients within the federation, while "i_max" denotes the client identified as the most distant from the others. Additionally, when defining the distance matrix Di,j , we use the same variable "i" to denote clients within the federation, emphasizing that this matrix captures the relationships between clients in the federation. To further enhance clarity, we will include comments in Algorithm 1, including one stating that "i" represents a client in the federation.
>
> Thank you once again for your valuable feedback, and we look forward to addressing any further questions or concerns you may have.

---

### Meta-Review · Area_Chair_YH9e · 2024-04-04

**Recommendation:** Accept (Poster)
**Confidence:** 4

**Metareview:**

The reviewers emphasized the novelty of the method and the quality of the experimental evaluation. The authors engaged in the rebuttal process.

---

### Decision · Program_Chairs · 2024-04-06

Accept (Poster)